Methods

# Identification of single-cell blasts in pediatric acute myeloid leukemia using an autoencoder

Alice Driessen[1,2,*], Susanne Unger[3,*], An-phi Nguyen[1], Rhonda E Ries[4], Soheil Meshinchi[4], Stefanie Kreutmair[3,5], Chiara Alberti[3], Pavel Sumazin[6], Richard Aplenc[7], Michele S Redell[8], Burkhard Becher[3,†], María Rodríguez Martínez[1,†]

**Pediatric acute myeloid leukemia (AML) is an aggressive blood cancer with a poor prognosis and high relapse rate. Current challenges in the identification of immunotherapy targets arise from patient-specific blast immunophenotypes and their change during disease progression. To overcome this, we present a new computational research tool to rapidly identify malignant cells. We generated single-cell flow cytometry profiles of 21 pediatric AML patients with matched samples at diagnosis, remission, and relapse. We coupled a classifier to an autoencoder for anomaly detection and classified malignant blasts with 90% accuracy. Moreover, our method assigns a developmental stage to blasts at the single-cell level, improving current classification approaches based on differentiation of the dominant phenotype. We observed major immunophenotype and developmental stage alterations between diagnosis and relapse. Patients with KMT2A rearrangement had more profound changes in their blast immunophenotypes at relapse compared to patients with other molecular features. Our method provides new insights into the immunophenotypic composition of AML blasts in an unbiased fashion and can help to define immunotherapy targets that might improve personalized AML treatment.**

## Introduction

Acute myeloid leukemia (AML) is a hematological cancer characterized by the accumulation and expansion of immature cells of the myeloid lineage in the bone marrow. The standard treatment of AML is chemotherapy, followed by stem cell transplantation during the initial remission phase for patients at high risk of relapse. Despite aggressive treatment, nearly 40% of pediatric patients eventually relapse (Aplenc et al, 2020). Once relapse occurs, the disease often becomes resistant to chemotherapy and a cure is hard to achieve (Hoffman et al, 2021).

Recent progress in chemotherapy has stagnated, as increasing treatment intensity has only resulted in elevated toxicity without improving clinical outcomes (Rubnitz et al, 2010; Elgarten et al, 2021). Conversely, targeted therapies, like antibody immunotherapies, have demonstrated success in the treatment of pediatric leukemias (Brivio et al, 2022). Immunotherapies can target chemotherapy-resistant cells and achieve long-term remission. However, finding therapeutic targets for AML is challenging, as many of the surface proteins on AML cells are also expressed on healthy bone marrow cells. In addition, identifying suitable targets for AML therapy is hampered by the heterogeneity and complex clonal composition of the cancer, as well as its complex evolution as the disease progresses (Langebrake et al, 2005; Zhai et al, 2022). Therefore, there is a pressing need to identify phenotypic characteristics specific to chemotherapy-resistant AML cells. These features may facilitate risk assessment and suggest potential targets for the development of immunotherapies.

Currently, the identification of malignant cells depends on manual annotations. Pathologists employ histological slide analysis from a few cells or flow cytometry to identify leukemia-associated immunophenotypes (LAIP) with a limited set of markers. Here, we present a machine-learning approach for research purposes to scale this process to efficiently manage larger numbers of cells, samples, and markers. We used anomaly detection to identify aberrant immunophenotypes that significantly deviate from the phenotype of healthy developing bone marrow cells. Anomaly detection is a machine-learning approach that seeks to identify patterns and occurrences that stand out from the expected behavior (Chandola et al, 2009).

Autoencoders provide a well-established method for anomaly detection (An & Cho, 2015; Wagner et al, 2019). Autoencoders are

---

[1]Data and AI Research, IBM Research Europe, Zürich, Switzerland   [2]ETH Zürich, Zürich, Switzerland   [3]Institute of Experimental Immunology, University of Zurich, Zurich, Switzerland   [4]Fred Hutchinson Cancer Research Center, Seattle, WA, USA   [5]Department of Medical Oncology and Hematology, University Hospital Zürich, Zürich, Switzerland   [6]Department of Pediatrics, Baylor College of Medicine, Houston, TX, USA   [7]Children's Hospital of Philadelphia, Philadelphia, PA, USA   [8]Texas Children's Cancer and Hematology Center, Baylor College of Medicine, Houston, TX, USA

Correspondence: becher@immunology.uzh.ch; maria.rodriguezmartinez@yale.edu
María Rodríguez Martínez's present address is Department of Biomedical Informatics and Data Science at Yale School of Medicine, New Haven, CT, USA
*Alice Driessen and Susanne Unger contributed equally to this work
†Burkhard Becher and María Rodríguez Martínez are joint senior authors

---

simple neural networks optimized to reconstruct input after passing it through a lower dimensional bottleneck. Specifically, an encoder projects input samples into the lower dimensional latent space and a decoder reverses this process to reconstruct the original inputs. The autoencoder is optimized to minimize the reconstruction error between the input and output. Once trained, the latent dimension offers a low-dimensional representation of the samples, which can be used to visualize the developmental trajectory of bone marrow cells. For anomaly detection, we trained the network exclusively on remission cells (the reference class) and asked the network to reconstruct diagnosis and relapse cells (a mixture of healthy and cancer cells). Because the network has not seen cancer cells (the anomaly class), we expect the model to reconstruct their aberrant immunophenotypes with a higher error. The difference in the reconstruction error between cancer and healthy cells can be used to distinguish between normal and malignant cells.

We used high-dimensional single-cell cytometry data to characterize bone marrow cells from pediatric AML patients collected at diagnosis, remission, and relapse. We show that our approach yields high accuracy and use it to classify all cells in our cohort. Moreover, the latent space of the autoencoder provides a reference single-cell developmental map of healthy bone marrow, onto which we can project new cells. This developmental map helps us determine the developmental stage of AML samples and visualize the cellular and phenotypic differences between diagnosis and relapse.

# Results

## Study participants and experimental approach

To investigate AML blasts and phenotypic differences between diagnosis and relapse, we performed spectral flow cytometry of 20 pediatric AML patients' matched diagnosis and relapse bone marrow samples (Fig 1A). Samples collected at the end of induction 1 chemotherapy (EOI1) were obtained from four of these patients and were considered remission samples. One additional patient contributed a remission sample only (Section 5.1).

We designed a spectral flow cytometry panel with 21 markers covering the major physiological developmental stages and cell subsets of the myeloid lineage, and lineage markers for T cells, B cells, and NK cells (Section 5.2). Because AML occurs in the myeloid lineage, we excluded all cells from the lymphoid lineage, plasmacytoid dendritic cells, and the non-malignant myeloid cell types, erythroblasts and basophils (Figs 1A and S1A–C). The remaining myeloid cells were labeled as hematopoietic stem and progenitor cells (HSPCs)/myeloid cells throughout the article (Figs 1B and S1B). Lymphoid cells, basophils, erythroblasts, and pDCs clustered by lineage, with extensive overlap across patients and samples and low individuality scores (Figs 1C and S1D, Section 5.4). Conversely, the HSPCs/myeloid cells showed great heterogeneity and high individuality scores across patients (Figs 1C and S1D), often forming distinct clusters when visualized with an UMAP (Figs 1D and S1E).

To train and validate our machine-learning models, we first manually annotated clusters of HSPCs/myeloid cells as either healthy or malignant. For this, we split the HSPC/myeloid cell raw data into two subsets, the remission cells (cells from EOI1) and non-remission cells (comprising diagnosis and relapse cells), and clustered each of them separately (Section 5.5). Because the remission samples contained ≤0.02% residual blasts, we assumed all remission cells to be healthy myeloid cells. We annotated clusters as HSPCs, monoblasts/myeloblasts, promonocytes, and monocytes based on established markers for hematopoietic differentiation (Figs 1E and F and S1F). To annotate the non-remission cells, we clustered each patient separately and considered clusters with deviations from normally occurring phenotypes during myelopoiesis as malignant (Figs 1G and S2, Section 5.5).

## Autoencoder coupled to a classifier successfully identifies malignant cells

We assumed that malignant cells from non-remission samples show an atypical immunophenotype detectable by machine learning. Hence, we used an autoencoder to identify these anomalous phenotypic patterns. We trained the autoencoder to accurately learn and reconstruct the marker expression of remission cells (Fig 2A, Section 5.6). Remission cells capturing the full age range of our patient were chosen as a reference dataset to avoid the confounding factors of age when using adult bone marrow. In addition, relapse samples ensure that detected aberrant expression patterns are not due to the chemotoxic effects of the treatment but reappearing AML blasts. Because the model is trained only on remission samples, we expect malignant cells to have a poorer reconstruction and thus higher reconstruction error. To label the aberrant cells, we trained a classifier that leveraged the squared reconstruction error (SRE) provided by the autoencoder—a 23-dimensional vector per cell—to identify blasts (Section 5.7).

Our autoencoder achieved a mean SRE of 0.006 over five cross-validation folds when reconstructing remission cells. Next, we used this autoencoder to generate the SRE per marker for the subset of annotated cells, which was used to train the classifier. Our random forest classifier achieved an average accuracy of 0.91 over all cross-validation folds (Platt, 2000). Finally, we used the trained autoencoder coupled with the classifier to identify malignant cells in the whole dataset. We achieved an accuracy of 0.90 for all annotated cells and a balanced accuracy of 0.82, demonstrating good performance across both classes (Fig S3C).

We investigated the non-remission cells, which were manually annotated as malignant but misclassified as healthy cells. Misclassified cells are spread out over the UMAP, but there are a few regions with a high density of misclassified blasts (Fig 2B). Most of these cells originated from four samples spanning four patients. For all other patients, less than 10% of the blasts were misclassified (Fig 2C).

Low CD45 expression is a typical LAIP and therefore often used for blast identification in conventional flow cytometry gating (Fig S3D). The CD45low gate achieves 0.78 accuracy (balanced accuracy 0.67; Fig S3E) on annotated cells, lower than our classifier. Among cells identified as blasts by our classifier, 82% are also in the CD45low gate. However, the CD45high gate underperforms in

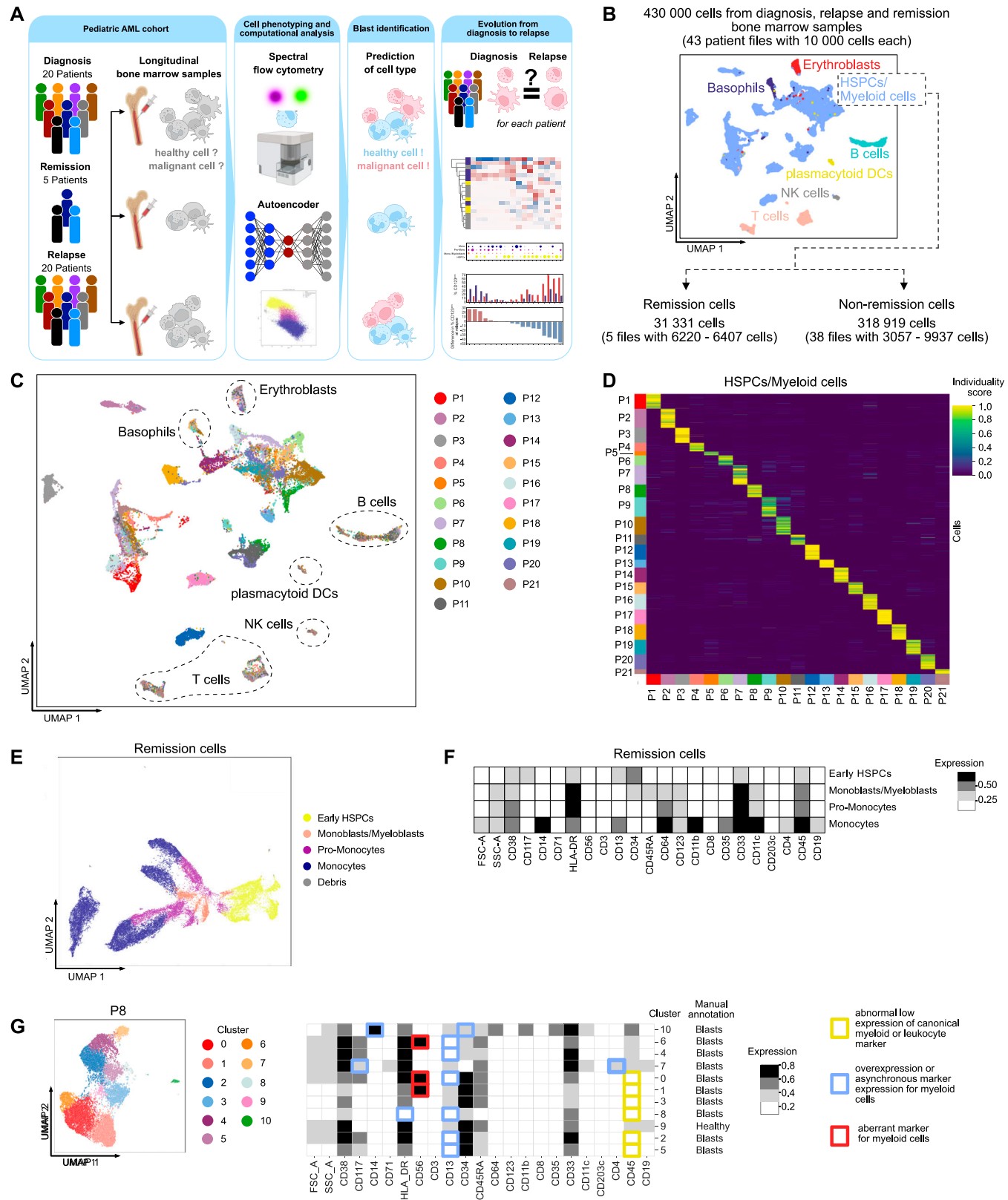

**Figure 1. Experimental approach and data processing.**
**(A)** Schematic overview of the workflow. **(B)** Selection of hematopoietic stem and progenitor cells (HSPCs)/myeloid cells from bone marrow samples. **(C)** Patient mixing in different bone marrow cell types and HSPCs/myeloid cells. **(D)** Individuality of HSPCs/myeloid cells between patients (see Section 5.4 for details on individuality).

detecting non-malignant cells, identifying only 52% of all cells manually annotated as healthy, and 50% of healthy cells predicted by our classifier (Fig 2D). The two methods also disagree on the fraction of blasts in some samples, particularly P1, P3, P7, and P12 (Fig 2E), where our method identifies a significantly higher blast fraction than the CD45low gate. This shows the superiority of our approach based on 23 inputs and the ability to learn a non-linear relationship instead of a one-dimensional CD45low gate with a linear threshold. For several samples, our prediction differed from blast counts of the clinical report (Fig S3F). However, this is not an appropriate comparison because the two methods differ in sample condition, processing, and blast assessment strategy. The clinical laboratory measured blasts in fresh samples, whereas we analyzed samples that had been ficolled, cryopreserved, and thawed. In addition, a blast count in the clinic relies on morphology and immunophenotyping with different marker combinations. Therefore, comparing our prediction of blast frequencies with diagnostic reports is not an appropriate evaluation metric.

Regarding the predictions of our classifier, detected blasts have a high reconstruction error for established AML blast markers, such as CD56, CD45RA, CD34, and CD33, indicating their abnormal expression (Fig 2F). Conversely, undetected malignant cells have a lower error, particularly for CD56, CD45RA, HLA-DR, and CD34 (Fig S3G). The mean-squared error is biased toward high expression values, so this low error could be due to low initial expression levels or a small deviation from normal expression levels. The reconstruction errors for all annotated cells can be seen in Fig S3G and H. In addition, we can see how important each marker is for blast classification. HLA-DR, CD33, CD34, and CD45RA are the topmost important markers for the classifier to distinguish blasts and healthy cells (Fig S3I).

### Autoencoder latent space can be used to detect the AML developmental stage

Autoencoders can map high-dimensional inputs into lower dimensional representations, in our case reducing from 23 to 4 dimensions in the latent space (Fig 3A). Remission cells mapped to the first two latent dimensions form a continuous trajectory from immature progenitors to mature monocytes, capturing the healthy development of bone marrow myeloid cells (Fig 3A, left panel).

First, we projected the predicted blasts of diagnosis samples onto the latent space. Blasts naturally fall onto the developmental trajectory, which allows us to visualize their developmental stage (Fig 3B). Other dimensionality reduction techniques such as the UMAP often separate diagnosis and remission cells. Similarly, a neural network trained to classify healthy and malignant cells directly from marker expression also segregated remission and malignant cells in the latent space, preventing the mapping of malignant cells onto healthy developmental trajectory (Figs 4A and S3A and B, Section 5.11).

Visual inspection of the latent space revealed that eight samples contained almost exclusively blasts with an HSPC-like phenotype,

whereas three others exhibited a monocyte-like phenotype (Fig 3B). Most samples showed a mixture of all phenotypes. To quantify the developmental composition of AML samples, we used a k-nearest-neighbor classifier in the latent space to assign a cell type to each predicted blast (Cover & Hart, 1967) (Section 5.8). We then analyzed the distribution of predicted cell types in each sample. Our developmental stage assignment matched the World Health Organization (WHO) classification (Fig 3C). For further validation, we examined the expression of known marker proteins (Fig S4B) and the Euclidian distance in the marker space between the predicted blast cell type and the same cell type in remission cells (Fig S4C), finding good agreement between both sets.

Similarly, we projected the relapse cells onto the latent space, predicted their developmental stage, and inspected phenotypic shifts from diagnosis to relapse. In certain patients, the cellular phenotype at relapse substantially differed from the diagnosis (Fig 3D). To investigate whether this shift was due to the expansion of therapy-resistant cells already present at diagnosis, we constructed a Gaussian mixture model and investigated the similarity between diagnosis and relapse cells (Pedregosa et al, 2011) (Section 5.9). For one patient with a particularly large compositional shift, the relapse phenotype was already observable at diagnosis (Fig 3E). This suggests that diagnosis cells that survive therapy might proliferate and contribute to the emergence of "new" relapse phenotypes. The cell-type composition of all samples can be seen in Fig 3F.

### KMT2A rearrangement is associated with an unstable immunophenotype

Given the strong differences between the diagnosis and relapse samples in some patients, we investigated marker expression changes, as this can help identify new immunotherapy targets. First, we calculated the percentage of cells positive for each marker per sample (Section 5.10, Fig 4A). Then, we counted the number of patients where the percentage of predicted blasts expressing each marker changed more than ±5% from diagnosis to relapse (Fig 4B). 11 markers were changed in at least 50% of the patients with matched diagnosis and relapse samples. CD123 and CD13 changed in 13 of the 18 (72%) patients. CD123 was lost in 9 and gained in 4 patients, whereas CD13 was lost in 4 and gained in 9 patients (Fig 4B and C). HLA-DR, CD38, and CD34 expression changed in 12 patients (67% of patients), and CD34 was gained in 7 patients (Fig 4B and D). CD117 and CD45RA expression varied in 11 patients (61%), whereas CD33, CD64, CD56, and CD11c changed in 9 patients (50% of patients). CD33, a target in current clinical trials, was gained in 6 and lost in 3 patients (Fig 4E). Interestingly, although CD56 expression changed in half of the patients, it was never completely lost in the relapse sample if it was present at diagnosis (Fig 4F). To better understand marker changes per patient, we calculated the change in the fraction of cells expressing each marker between diagnosis and relapse samples of each patient (Section 5.10). Patients with a KMT2A rearrangement exhibited greater phenotype instability,

---

**(E)** UMAP of remission HSPCs/myeloid cells, colored by the annotated cell type. **(F)** Median marker expression of remission cell types. **(G)** UMAP of non-remission HSPCs/myeloid cells of P8 and heatmap with the median marker expression of identified clusters. Aberrant marker expression in clusters is highlighted.

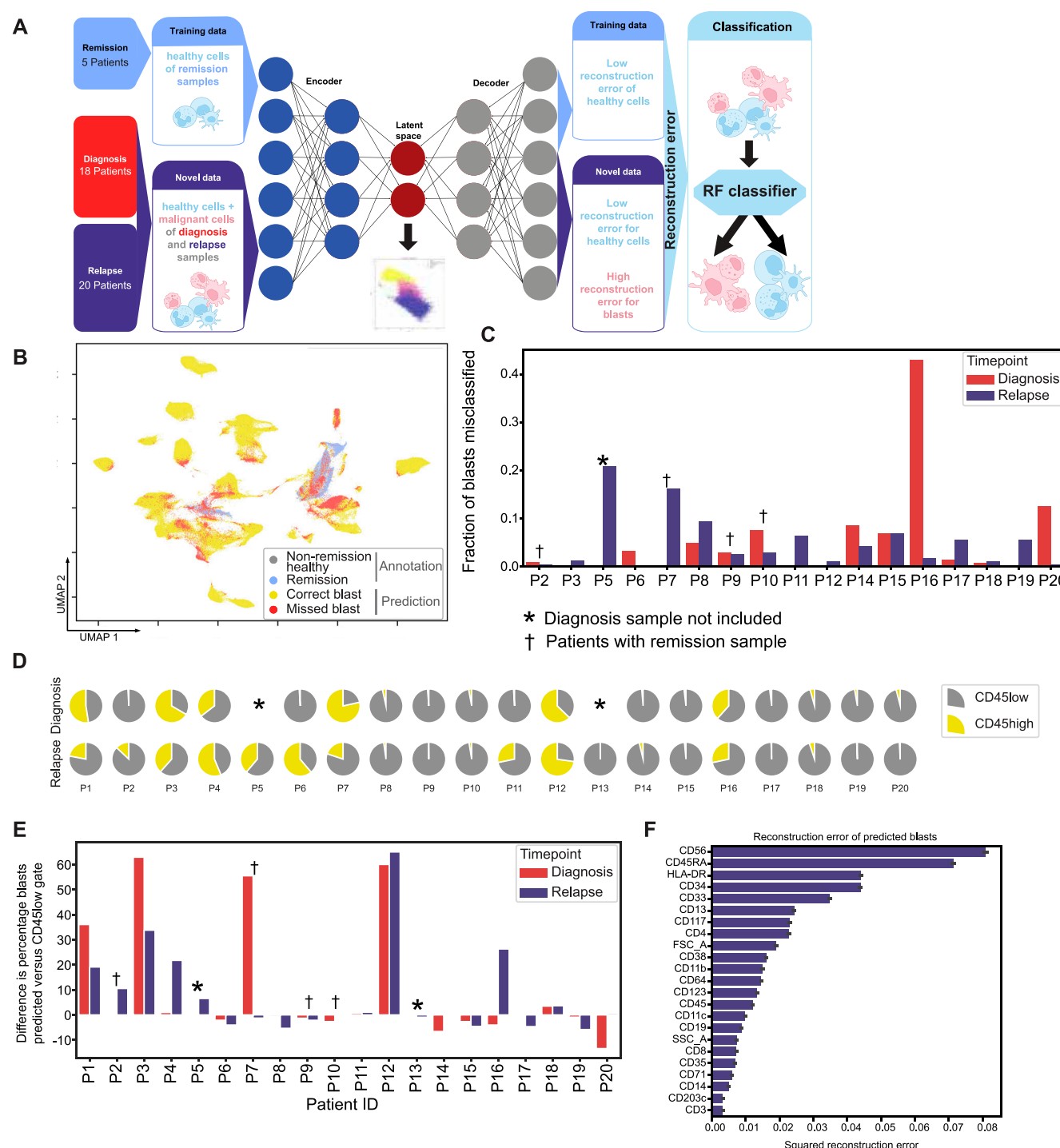

**Figure 2. Evaluation of single-cell blast prediction.**
**(A)** Schematic overview of the modeling pipeline. **(B)** Classifier predictions on annotated non-remission cells and remission cells on an UMAP of all hematopoietic stem and progenitor cells/myeloid cells. **(C)** Fraction of blast cells that are misclassified in each sample (the number of misclassified blasts divided by the number of predicted blasts in the sample). **(D)** Comparison of our modeling pipeline with CD45low gating. Full circles represent the blasts predicted by our classifier for each sample. The fraction of predicted blasts that are in the CD45low gate is shown in gray; highlighted in yellow is the fraction of predicted blasts that would have been missed by the CD45low gate. **(E)** Predicted percentage of blasts in the entire sample compared with the percentage of blasts in CD45low gate for the entire sample. **(F)** Reconstruction error for each marker for cells predicted to be blasts. Error bars indicate the 95% confidence interval.

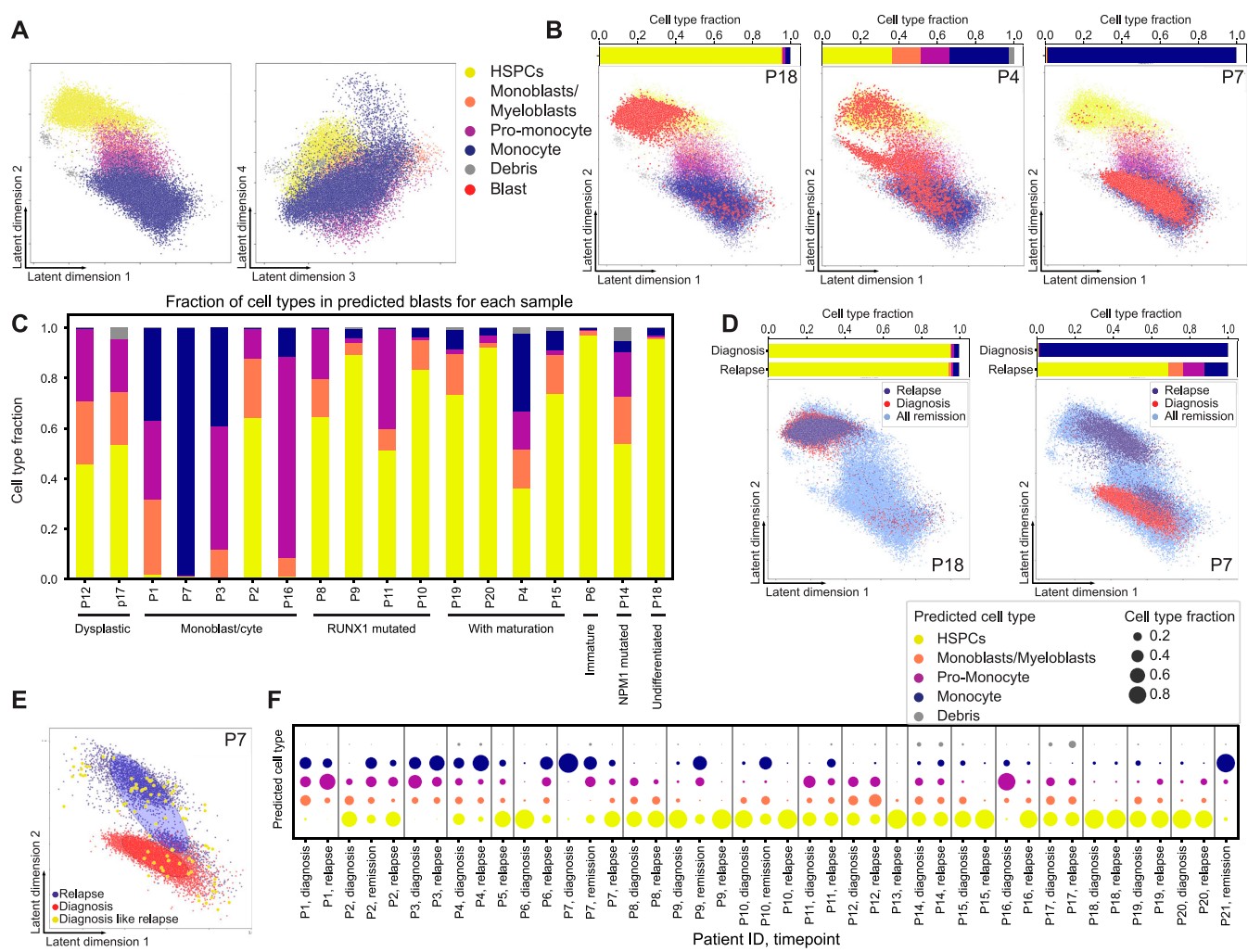

**Figure 3. Single-cell map in the latent space used for acute myeloid leukemia developmental stage detection.**
**(A)** Remission cells in the four dimensions of the autoencoder latent space. **(B)** Three examples of diagnosis samples with remission cells in the first two dimensions of the latent space. The top bars show the predicted fraction of each cell type in each of the samples. **(C)** Cell-type composition of diagnosis samples annotated with the World Health Organization labels from the diagnostic report. **(D)** Two examples of patients' diagnosis and relapse samples and all remission cells in the first two dimensions of the latent space. The top bars show the predicted cell-type composition of the patients' samples. **(E)** Patient's diagnosis and relapse samples in the first two latent dimensions. Ellipses depict covariances of the single-component Gaussian mixture models fitted to diagnosis and relapse separately. Highlighted are diagnosis cells with a higher log-likelihood under the relapse-Gaussian than under the diagnosis-Gaussian. **(F)** Predicted cell-type fractions of predicted blasts in each of the samples.

showing significant changes in CD64, CD11c, CD11b, CD14, CD35, CD4, and CD45, markers that remained stable in other patients (Fig 4G). Furthermore, we calculated the Euclidean distance between the fraction of cell types at diagnosis and relapse for each patient and found that patients with a KMT2A rearrangement have a larger change in phenotypic composition than patients with other molecular features (Fig 4H).

# Discussion

We presented a high-quality single-cell flow cytometry dataset with matched samples at diagnosis, relapse, and remission. Leveraging this dataset, we constructed a computational framework for the large-scale analysis of single cells, automatic blast detection, and

sample developmental stage characterization. The advantage of our method lies in the ability to rapidly classify thousands of single cells across multiple samples and patients using a data-driven methodology. We investigated the phenotypes of blasts detected with our model and analyzed immunophenotypic changes from diagnosis to relapse. Our study offers a research tool, which provides a comprehensive view of disease progression in pediatric AML. This deep and longitudinal characterization of lineage marker expression could be used as a research tool to retrospectively evaluate individualized targeted therapy decisions based on the presence or absence of markers on leukemic blasts.

Our model's ability to assign developmental stages to AML blasts has significant implications for clinical decision-making. A recent RNA sequencing–based study showed that the differentiation stage of blasts can predict response to chemotherapy and drug

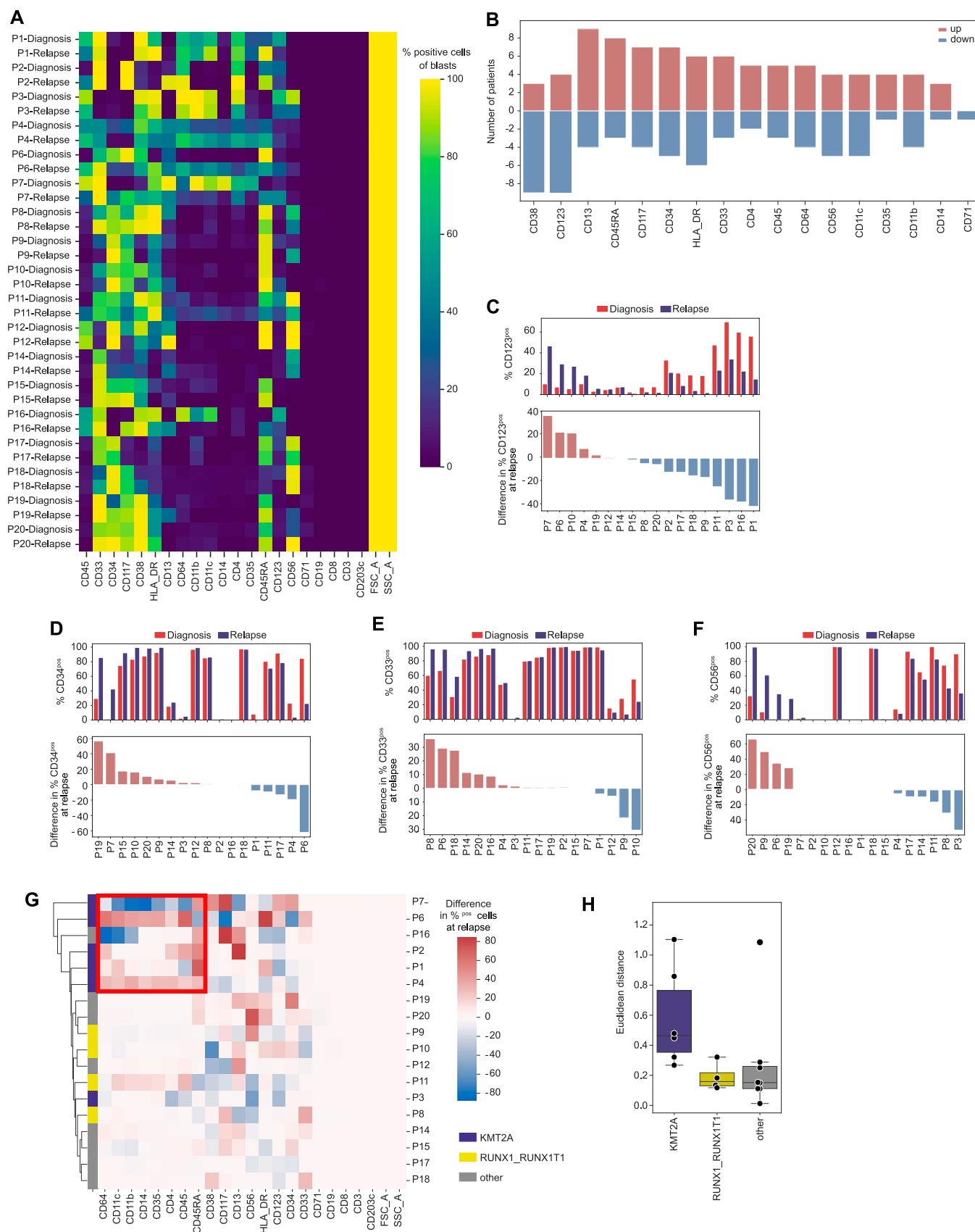

sensitivity profiles of targeted therapy (Zeng et al, 2022). In addition, adult patients with more differentiated monocytic cells often exhibit resistance to Venetoclax (Pei et al, 2020). It remains uncertain whether these findings generalize to pediatric cohorts, given the scarcity of studies (Masetti et al, 2023; Trabal et al, 2023). However, a small study involving five pediatric AML patients found that the two patients with M5 monocytic AML did not respond to Venetoclax, whereas those with M4 myelomonocytic leukemia achieved complete remission (Winters et al, 2020). Unfortunately, the developmental stage was not evaluated in the only two other available studies (Masetti et al, 2023; Trabal et al, 2023). Our method is superior to the traditional French–American–British (FAB) or WHO classification of AML maturation, as we can identify the developmental stages at the single-cell level. This provides a more fine-grained understanding of the cellular composition of AML, beyond simply considering the dominant phenotype (Zeng et al, 2022).

Some patients showed distinct phenotypes at diagnosis and relapse. No correlation was observed between these changes and the clinical data, potentially because of our small cohort size. It is still under debate whether new phenotypes emerge after chemotherapy or whether rare phenotypes already present at diagnosis expand at relapse. Our findings generally support the latter hypothesis. However, because of our study's limited sample size, the emergence of entirely new phenotypes in other patients cannot be excluded.

Immunophenotypic shifts from diagnosis to relapse have been previously reported in both adult and childhood AML; often, CD117, CD13, and CD34 are the most variable markers (Baer et al, 2001; Langebrake et al, 2005; Martínez-Losada et al, 2018). Our study confirmed these findings and discovered additional markers that significantly change in many patients, such as CD123 (in 72% of patients), CD38 (in 67%), and CD45RA (in 61%). However, it is challenging to compare results across reports, because of varying definitions of "change of phenotype." Our study exploited 23 parameters, increasing precision over previous studies constrained to 4–8 parameters. Using a small number of markers might compromise the distinction between blasts and healthy cells, resulting in over- or underestimation of marker changes. For instance, we demonstrated that using CD45low as the primary threshold to identify blasts could overlook 20–65% of blasts in 21% of the samples. Our model improves both the sensitivity and specificity of blast detection compared with the CD45low gating.

We also observed that 10 patients (55%) expressed CD56 at diagnosis, a marker associated with poor prognosis in childhood AML (Liang et al, 2022). Even though the frequency of cells expressing CD56 at relapse decreased in six of these patients compared with diagnosis, some blasts still tested positive for CD56. The role of CD56 in AML cells remains elusive. However, evidence suggests it might

be involved in the maintenance of leukemic stem cells, in agreement with our observations (Martínez-Losada et al, 2018; Sasca et al, 2019).

Markers gained over time or not lost on blasts, such as CD56, can be potential targets for AML immunotherapy, as their expression at relapse makes them suitable to target therapy-resistant cells. Conversely, markers lost over time are likely poor targets. CD123 and CD33 are currently included in many clinical trials for AML immunotherapy (Chen et al, 2023). However, CD123 was less frequent at relapse compared with diagnosis in 50% of our patients, even in the absence of selection pressure for antigen escape because of prior immunotherapies. Contrarily, CD33 shows more promise, being lost only in 3 patients (17%) and gained in 6 (33%). Similarly, the fraction of CD34$^+$ cells increase in relapse for many patients. However, despite its potential as a therapeutic target, there is, to our knowledge, currently no CD34-targeting immunotherapy in development. Our cohort is a small sample of the real-world pediatric AML population, with minor differences in age, sex, and mutational frequency compared with global frequencies (Hossain & Xie, 2015; Conneely & Rau, 2020). Therefore, these markers should be validated in other cohorts to gain a representative insight into the therapeutic relevance of these markers.

We observed major immunophenotype and developmental stage alterations between diagnosis and relapse, confirming findings in other studies (Langebrake et al, 2005; Bachas et al, 2015; Konoplev et al, 2022). Studies on pediatric AML typically report a shift toward a more immature phenotype at relapse (Langebrake et al, 2005; Bachas et al, 2015). Interestingly, we observe that most patients with KMT2A rearrangements show a more mature immunophenotype at relapse, in line with reports of adult AML that shifts in relapse can occur either to more mature or to more immature (Konoplev et al, 2022). Although we do not address the biological effect of phenotypic changes in KMT2A-rearranged cases in this article, KTM2A-rearranged samples could be tested for more markers than the traditional LAIP profiles, to validate this observation in more samples. Studies with high-resolution measurements of KMT2A-rearranged cells could also further investigate the effects or causes of the phenotypic shift.

Our workflow has not been tested on external samples because of the difficulty of finding independent datasets with matching marker panels for validation. Although our trained model may not be directly applicable to new datasets, the computational pipeline itself can be easily adapted to accommodate different marker panels. An additional limitation is the lack of remission samples for most patients. Including additional remission samples during the training of the autoencoder will enhance the representation quality of healthy cells, thereby improving the robustness of blast detection.

**Figure 4. Differences in blast phenotypes at relapse.**
**(A)** Percentage of predictive blasts that are positive for a marker in each sample. Marker thresholds are listed in Table S5. **(B)** Number of patients with a marker change of at least 5% up or down for each marker. **(C, D, E, F)** Difference in the percentage of positive cells for indicated markers in predicted relapse blasts compared with diagnosis blasts. **(C, D, E, F)** Percentage of positive cells for CD123 (C), CD34 (D), CD33 (E), and CD56 (F) in predicted diagnosis and relapse blasts (upper graphs) and the difference at relapse compared with diagnosis (bottom graphs). **(A, C, D, G)** Marker changes between diagnosis and relapse as shown in panels (A, C, D) per patient, clustered for patients and markers. Highlighted are markers changing for patients with KMT2A mutation but not in other patients. **(H)** Euclidean distance between the predicted cell-type vectors for each sample split and colored by mutation status. The cell-type vectors are the fractions of predicted cell types (see Fig 3F for the blasts in a sample). The line indicates the median; the box, the interquartile range; and the whiskers, 1.5 times the interquartile range.

Despite these limitations, the classifier trained on the autoencoder reconstruction error demonstrated excellent performance in blast classification and developmental stage identification. This suggests that the autoencoder effectively captured the underlying patterns associated with myeloid and progenitor cells. This workflow could also be adapted to other single-cell omics technologies. A similar approach has been used to assign a phenotypic anomaly score to breast cancer cells in single-cell mass cytometry data (Wagner et al, 2019). In RNA sequencing–based methods, anomaly detection has been applied to identify rare cell subpopulations, rather than malignant cell detection (Feng et al, 2022; Mallick et al, 2023). However, we recommend screening for genetic abnormalities to identify malignant cells when genetic information is available (van Galen et al, 2019). Overall, our findings are promising for the reusability and broader applicability of our workflow.

This work has improved the detection of single-cell blasts in blood malignancies. To further expand on this framework, the marker panel could be expanded to increase sensitivity. In addition, computational resources open the possibility to set up a similar framework that could work with different datasets. Given many datasets with a valid reference and disease samples, a powerful model could learn to mitigate batch effects. The model would learn to distinguish the malignant versus healthy signal from the batch and patient signal, if trained on sufficient high-quality data. This would mean that MRD detection or developmental stage composition of unseen samples could be predicted in the clinic. Therefore, we call for future studies to adopt larger marker panels, also include remission samples, and make data available whenever possible. Then, AI-enabled MRD detection is on the horizon.

# Materials and Methods

### Study participants' details

Cryopreserved bone marrow specimens from 20 pediatric patients with de novo AML were obtained from the Children's Oncology Group (COG) Biopathology Center. All patients were enrolled on the phase III treatment protocol AAML1031 and were randomized to the standard arm (Arm A). The treatment regimens and outcomes for this trial have been reported (Aplenc et al, 2020). Sample selection criteria included the following: diagnosis/relapse pairs from patients enrolled on AAML1031, Arm A, who consented to banking and who had three or more vials of bone marrow (not peripheral blood) banked at both timepoints. The requirement for multiple vials is imposed by the NCI to avoid depletion of the holdings for any individual patient and timepoint. In addition, the vials should contain at least a certain number of cells.

All patients contributed a sample from diagnosis and relapse. Four of the patients also contributed a remission sample at the end of induction 1 chemotherapy (EOI1). Clinical flow cytometry reported residual blasts in only one of these samples (0.02%). One additional remission sample without residual AML at EOI1 was included from a non-COG patient who was not enrolled in the AAML1031 study but

followed the same regimen. The National Cancer Institute's Central Institutional Review Board (IRB) and IRB at each enrolling center approved the study, and patient families gave written informed consent, in accordance with the Declaration of Helsinki, to allow the bone marrow samples to be banked for future research. Bone marrow samples were enriched for mononuclear cells by density centrifugation and cryopreserved. Patient demographic and clinical information was provided by the COG Statistics and Data Center (Tables S1, S2, and S3). The cohort shows minor differences in sex, **Q:2** age, and mutational frequency compared with global pediatric AML frequencies. These differences are due to sample selection based on the above-mentioned criteria. The incidence in males and females is equal in global pediatric AML statistics (Hossain & Xie, 2015), whereas the cohort contains 62% males (13) and 38% females (8). In addition, we have an overrepresentation of KMT2A rearrangements (Conneely & Rau, 2020).

### Spectral flow cytometry

Frozen bone marrow samples were thawed by centrifugation using cryo–thaw devices (Medax) and collected in cell culture medium (RPMI 1640, 10% FBS [Gibco], 1 × GlutaMAX [Gibco], and 1 × penicillin–streptomycin [Gibco]) supplemented with 5 U/ml Benzonase. Cells were spun down and resuspended in cell culture medium supplemented with 2 U/ml Benzonase.

For the staining with fluorophore-coupled antibodies, $8 \times 10^5$ cells were transferred into a 96-well V-bottom plate, spun down, resuspended in 50 µl of Zombie NIR Fixable Viability Dye (1:500; BioLegend) for 15 min, and washed with PBS with 0.5% BSA. 50 µl of FcX Fc receptor block (1:200; BioLegend) was added for 15 min, followed by another washing step. Subsequently, cells were incubated with 50 µl of a flow cytometry antibody mixture for 15 min. After washing, samples were fixed using 50 µl of 2% PFA for 15 min, washed with PBS, and analyzed on Cytek Aurora 5L Spectral Analyzer. All incubation and washing steps during the staining procedure were performed at 4°C.

The initial flow cytometry panel consisted of 26 markers: CD45, CD19, CD3, CD4, CD8, CD56, CD34, CD45RA, HLA-DR, CD117, CD38, CD33, CD35, CD13, CD123, CD64, CD11b, CD11c, CD14, CD203c, CD71, CD42b, CD16, CD300e, CD112, and CD155. We ignored CD42b for the analysis because the signal for CD42b was due to platelet aggregates on cells and did not detect megakaryocytes specifically. We excluded CD16, CD300e, CD112, and CD155 entirely because of staining artifacts. We selected 21 markers from the flow cytometry panel and included CD45, CD19, CD3, CD4, CD8, and CD56 to distinguish the lymphoid and myeloid lineages. For a more fine-grained insight into the myeloid lineage, we included CD34, CD45RA, HLA-DR, CD117, CD38, CD33, CD35, CD13, CD123, CD64, CD11b, CD11c, and CD14 for granulocytes, monocytes, and dendritic cell development. We included CD203c for basophil identification, and CD71 for erythroblast identification. We also measured the forward and side scatter of the cells, which provided information about the cells' size and internal complexity. Finally, we used this 23-dimensional dataset for downstream analysis (antibodies are listed in Table S4). We excluded two samples from the analysis because of low cell counts (<10,000 cells).

## Data processing

FCS files were loaded into FlowJo (FlowJo Software, 2023) (Tree Star), and compensation of the flow cytometry data was manually adjusted. Two low-quality samples (Section 5.2) were excluded because of low cell counts (FlowJo Software, 2023). After exclusion of cell doublets and dead cells (for this gating strategy, see Fig S1A), 10,000 random cells per file were exported. The data were transformed with an inverse hyperbolic sine function (cofactor range 500–9,030, Table S5), typically used to manage the wide dynamic range of flow cytometry data. These cofactors were manually determined for each marker, to achieve a distribution with two clear peaks with the negative peak being around zero (den Braanker et al, 2021). Cofactors remained constant for all data subsets. Second, each marker was scaled between 0 and 1 using MinMaxScaling (Section 5.12). The data before these preprocessing steps are considered raw data. After the preprocessing, the data were clustered using FlowSOM (Van Gassen et al, 2015) into 80 clusters. Based on the expression of specific markers in the clusters, we merged some of them into the main lymphocyte and myeloid lineages and subsequently excluded B cells, T cells, NK cells, erythroblasts, plasmacytoid dendritic cells, and basophils. Neutrophils were not observed as the samples were enriched for mononuclear cells before cryopreservation. All remaining clusters expressed either CD45, CD34, CD33, CD64, HLA-DR, or CD71 and comprised either healthy HSPCs, healthy myeloid cells, or malignant AML blasts. These 350'250 cells are collectively referred to as HSPCs/myeloid cells from this point forward and used for downstream analyses.

Q:3

## Individuality

In our preliminary analyses, we noticed that patients and timepoints had quite distinct marker expression profiles. To quantify the degree of similarity across samples, patients, or timepoints, we calculated an *individuality score* (Wagner et al, 2019). This metric evaluates the class distribution of a cell's nearest neighbors. If most of the neighbors fall in the same class as the cell investigated, then the cell resembles other cells of its own class. In this case, the class is self-contained. Conversely, if the neighboring classes are evenly represented, there is class mixing and the cell exhibits phenotypic similarities with multiple classes, reducing its class-specific uniqueness. We used the *k*NN (Cover & Hart, 1967) method from *sklearn* in combination with posterior probability analyses to determine how self-contained the classes are. The posterior probability that a cell belongs to a class *i* given its *k*-nearest neighbors is given by

Q:4

$$p(class\_i | k\_NN) = \frac{\frac{n\_NN_{c=i}}{n\_class\_i}}{\sum_{i=1}^{C} \frac{1}{n\_class\_i}}.$$

In the above equation, $k\_NN$ are the $k$-nearest neighbors of a cell, and $n\_NN_{c=i}$ are the number of nearest neighbors belonging to class $i$. $n\_class\_i$ is the number of cells that belong to class $i$ in the data. $C$ is the total number of classes. This probability was calculated for each class and each cell.

## Single-cell annotation

We split the HSPC/myeloid cell raw data into subsets: remission cells and per-patient non-remission cells (comprising diagnosis and relapse cells). The MinMaxScaling was applied to each subset independently. Each subset was clustered with Pheno-Graph (Levine et al, 2015) using different numbers of nearest neighbors (k parameter). For each k value, the clustering was repeated 10 times with different random initializations. The adjusted rand index (ARI [Rand, 1971]) between all pairs of clustering was calculated, and the clustering with the highest average ARI was selected. 19 clusters for the remission samples were obtained and annotated based on established markers for hematopoietic differentiation such as CD33, CD34, CD117, CD64, CD11b, CD11c, and CD14 (Porwit & Béné, 2018). This allowed us to merge the 19 clusters into four subsets, namely, early HSPC, monoblasts/myeloblast, promonocyte, or monocyte. The low expression signal for CD19 expression on monocytes was due to background staining of the antibody and not considered aberrant. Three clusters were considered as cell debris because of expression profiles, which are not present in healthy bone marrow. For the annotation of non-remission HSPC/myeloid cells, all patients were clustered separately to achieve a high resolution for blast annotation. Clusters with aberrant expression profiles of normal myeloid differentiation with at least one of the following characteristics were considered blasts: (a) the aberrant expression of the NK cell marker CD56, (b) an abnormal low expression of a canonical myeloid (CD33) or leukocyte marker (CD45), and (c) the overexpression or asynchronous marker expression for myeloid cells, for example, a cluster with the overexpression of a precursor marker CD117 with an absence of CD34, but the overexpression of CD4 (Porwit & Béné, 2018).

## Autoencoder training

The raw HSPC/myeloid remission cells were split into a test and training set. The training set was further split into five cross-validation (CV) folds, yielding five training and validation sets. For each trained model, the training data were preprocessed as described in "Data processing." The test and validation data were transformed with the same cofactors and min-max–scaled with the minimum and maximum of the training data. The test data were not used for the autoencoder training but for the training of the classifier (Section 5.7).

Autoencoders were implemented in Python using the PyTorch (Paszke et al, 2019) and PyTorch Lightning (Falcon et al, 2020) libraries. We tested both autoencoder and variational autoencoder architectures with different hyperparameter choices (Table S6). We randomly selected 100 hyperparameter configurations and trained both model architectures on each of the five cross-validation folds, resulting in a total of 1,000 tested models. All models were trained to reconstruct the 23 flow cytometry inputs (Section 5.2) using a mean-squared error (MSE) loss. The variational autoencoder used an additional Kullback–Leibler loss (Kingma & Welling, 2013 *Preprint*). The model with the lowest average MSE over all 5 CV folds was retrained on the full training set and used for further analyses.

## Classification of malignant cells

We trained binary classifiers using the autoencoder's reconstruction error for each marker; that is, the input of the classifier was a 23-dimensional input vector containing each marker's error. As described in Section 5.6, the remission test data (i.e., data not used to train the autoencoder) were used as training data for the classifier. These data were combined with the manually annotated malignant cells from non-remission samples to create leave-one-out cross-validation splits. In each fold, annotated malignant cells (blasts) of one patient were set aside for validation. The size of the excluded malignant cluster was subsampled to match the size of the validation fold of the remission cells (1/20 of the remission test set). The remaining malignant cells served as the training set for that fold and were also subsampled to match the size of the remission training set (19/20 of the remission test set). This resulted in 20 balanced training and validation splits. For each split, the data were preprocessed with the same scaling factors used in the best-performing autoencoder (Section 5.3).

We trained 100 models using these cross-validation splits and selected the model with the highest average validation accuracy over all folds. The selected model was then retrained on all folds, maintaining the balance in the data. The models and parameters explored are listed in Table S7. All models were implemented with the Scikit-learn (*sklearn*; Pedregosa et al, 2011) library in Python. For any parameter not listed, we used the default *sklearn* values.

### Cell-type prediction

We used the *sklearn* (Pedregosa et al, 2011) k-nearest neighbor (Cover & Hart, 1967) classifier to predict cell types of the non-remission samples based on their latent space encoding. The kNN classifier was trained using the latent encoding of the remission cells and used to classify the non-remission cells. We used the default *sklearn* parameters (*n_nearest_neighbors = 5*) and only considered the remission as potential neighbors in the classification task.

### Surviving cell analysis

We used a Gaussian mixture model (GMM) to investigate whether diagnosis cells that survive therapy might give rise to new relapse phenotypes; namely, using *sklearn* we fitted two separate GMMs with a single component each to the encoded diagnosis and relapse cells (i.e., diagnosis-GMM and relapse-GMM, respectively) (Pedregosa et al, 2011). For each cell, we evaluated the log-likelihood under both GMMs. Cells present at diagnosis that had higher log-likelihood under the relapse-GMM were labeled "Diagnosis like relapse." Similarly, relapse cells with higher log-likelihood under the diagnosis-GMM were labeled "Relapse-like diagnosis."

### Changes in the fraction of positive cells

To investigate the changes in marker expression between diagnosis and relapse blasts, we measured the fraction of cells positively stained for each marker. For the cells predicted by our classifier to be blasts (Section 5.7), we used marker-specific thresholds to determine which cells were positive for each marker. These thresholds were based on naturally occurring internal control cells with negative expression in the remission cells (Table S5). We calculated the fraction of predicted blasts that were positive for each marker and compared this fraction between diagnosis and relapse samples of the same patient (18 patients).

### Additional models trained directly on expression

We compared our computational workflow with models trained directly on the marker intensities. We trained these models following the same cross-validation scheme and hyperparameter search as described in Section 5.7 with three major differences. (1) The input was the marker expression/intensities instead of the reconstruction error. (2) Because there was no autoencoder used, we did not use the scaler used in the autoencoder training. Instead, we used the minimum and maximum of each train fold of the cross-validation to scale both the training and validation sets of that cross-validation fold with min-max scaling. (3) We trained extra multilayer perceptron (MLP) models, because we want to compare the latent space of the MLP with that of the autoencoder. However, the initial architectures of the MLPs had hidden layer dimensions ranging from 10 to 50 and the autoencoder has a latent dimension of 4. Therefore, we trained 50 extra models with parameter "hidden_layer_sizes" = [10, 4], resulting in a 4-dimensional layer comparably to the autoencoder. We randomly sampled the hyperparameters "activation function" and "solver."

We evaluated the accuracy of these models in single-cell blast classification on the remission cells and annotated blasts (Section 5.5, Fig S3A and B). For the comparison of the latent space, we investigated the latent space of the MLP with the highest accuracy overall and the highest accuracy of the MLPs with the "hidden_layer_sizes" = [10, 4]. For the overall best model, the hidden layers were as follows: [40, 60, 40, 10] and we used the UMAP to map the activation of each of these layers to a 2D space. We observed complete separation between the remission and malignant cells. The remission cells did form a developmental trajectory, but the malignant cells formed separate clusters outside of this trajectory. For the smaller MLP, we looked at the activations of the last, four-dimensional layer. Fig S4A shows the first two dimensions. We also investigated the third and fourth dimensions, as well as an UMAP of all four dimensions. In all cases, we observed a separation of the remission and malignant cells.

### MinMaxScaling

Min-max scaling scales all values between 0 and 1. The equation is as follows:

$$X_{min\_max\_scaled} = \frac{X - min(X)}{max(X) - min(X)}$$

We performed this scaling for each marker separately. Thus, to scale for *n* cells the marker *m*, *min(X)* is the minimum value among the *n* cells for marker *m*. *Max(X)* is the maximum value of marker *m*

among the *n* cells. *X* is the value of marker *m* in a given cell, and this formula is applied to each cell and each marker.

## Data and Code Availability

Q:5
Q:6
Spectral flow cytometry data have been deposited at (Unger, 2023) and is publicly available. All code has been deposited at https://github.com/DriessenA/auto_encoder_for_blast_detection and is publicly available.

## Supplementary Information

## Acknowledgements

Q:7
The present study was supported by the European Union's Horizon 2020 program (826121, iPC project), to S Unger, A-p Nguyen, P Sumazin, and M Rodríguez Martínez. A Driessen and M Rodríguez Martínez received funding from the European Union's Horizon 2020 research and innovation program under the Marie Skłodowska-Curie grant agreement no. 955321. The COG Biospecimen Bank is supported by funding from the NIH (U24 CA196173) and the St. Baldrick's Foundation. Funding for the COG Operations Center and the COG Statistics and Data Center is provided by the NIH (NCTN Operations Center Grant U10CA180886 and NCTN Statistics & Data Center Grant U10CA180899, respectively). The content is solely the responsibility of the authors and does not necessarily represent the official views of the National Institutes of Health. This project received funding from the European Research Council (ERC) under the European Union's Horizon 2020 research and innovation program grant agreement no. 882424 and the Swiss National Science Foundation (733 310030_170320, 310030_188450 and CRSII5_183478) to B Becher. M Rodríguez Martínez received funding from the Swiss National Science Foundation (CRSII5_193832, 200021_192128). S Kreutmair is a recipient of a research fellowship (442457282) of the German Research Foundation (DFG). We would like to acknowledge Marianna Rapsomaniki, who helped shaping the project at an early stage and provided feedback throughout the project.

### Author Contributions

A Driessen: conceptualization, data curation, formal analysis, validation, investigation, methodology, and writing—original draft, review, and editing.
S Unger: conceptualization, data curation, formal analysis, investigation, visualization, methodology, and writing—original draft, review, and editing.
A-p Nguyen: methodology.
RE Ries: resources and writing—review and editing.
S Meshinchi: resources and writing—review and editing.
S Kreutmair: investigation and writing—review and editing.
C Alberti: investigation.
P Sumazin: conceptualization, resources, funding acquisition, and writing—review and editing.
R Aplenc: conceptualization, resources, and writing—review and editing.
MS Redell: conceptualization, resources, data curation, supervision, funding acquisition, and writing—original draft, review, and editing.
B Becher: conceptualization, resources, supervision, funding acquisition, and writing—review and editing.
M Rodríguez Martínez: conceptualization, supervision, funding acquisition, and writing—review and editing.

### Conflict of Interest Statement

The authors declare that they have no conflict of interest.

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
