## [Reviewer comments · Life Science Alliance]

Life Science Alliance

Identification of single cell blasts in pediatric Acute Myeloid Leukemia using an Autoencoder

Alice Driessen, Susanne Unger, An-phi Nguyen, Rhonda Ries, Soheil Meshinchi, Stefanie Kreutmair, Chiara Alberti, Pavel Sumazin, Richard Aplenc, Michele Redell, Burkhard Becher, and María Rodríguez Martínez

DOI: <https://doi.org/10.26508/lsa.202402674>

Corresponding author(s): *María Rodríguez Martínez, Yale School of Medicine and Burkhard Becher, University of Zurich*

Review Timeline:

Submission Date:	2024-02-23
Editorial Decision:	2024-04-08
Revision Received:	2024-07-04
Editorial Decision:	2024-08-05
Revision Received:	2024-08-09
Accepted:	2024-08-19

Transaction Report:

April 8, 2024

Re: Life Science Alliance manuscript #LSA-2024-02674

Ms. Alice Driessen
IBM Research - Zurich
Säumerstrasse 4
Rüschlikon 8803
Switzerland

Dear Dr. Driessen,

Thank you for submitting your manuscript entitled "Identification of single cell blasts in pediatric Acute Myeloid Leukemia using an Autoencoder" to Life Science Alliance. The manuscript was assessed by expert reviewers, whose comments are appended to this letter. We invite you to submit a revised manuscript addressing the Reviewer comments.

Thank you for this interesting contribution to Life Science Alliance. We are looking forward to receiving your revised manuscript.

Sincerely,

B. MANUSCRIPT ORGANIZATION AND FORMATTING:

Reviewer #1 (Comments to the Authors (Required)):

Driessen and Unger et al. performed spectral flow cytometry on diagnosis, remission, and relapse samples from pediatric AML patients. To identify and quantify blasts, they implement an autoencoder trained on remission samples and pair it with a classifier trained on remission samples and annotated malignant cells from diagnosis and relapse samples. They also explore the utility of the autoencoder's latent dimensions to characterize the developmental stage of blasts and show correspondence with patients' clinical phenotype. Finally, they dissect changes in blast marker expression from diagnosis to relapse to offer insights about the appropriateness of specific therapeutic targets. Overall, the study generates insightful data and an innovative computational approach.

Major point:

1. The authors annotate only 16 of 37 non-remission HSPC/myeloid clusters as blasts, labeling only clusters with clear deviations in cluster marker expression from healthy cells. This makes evaluation of the classifier accuracy difficult, as only a fraction of the data has ground truth labels, and that fraction is the most easily identified malignant cells. The authors should use additional tools to annotate a great proportion of these data. The clinical blast counts should give a ballpark estimate of blast numbers in each patient, which could help guide annotation. Further, patient-specific clustering and patient-specific UMAP visualization often enable clearer segregation of healthy and malignant populations, which should enable annotation of more cells. Expanding the training set to include malignant cells with less obvious aberrant phenotypes may improve overall classifier performance and enable more insights in blast characterization in later figures.

Minor points:

1. Could the authors discuss if this analytic framework would be suitable using healthy control samples rather than remission samples as training data? While adding additional samples may be beyond the scope of a revision, it would be useful to have some discussion on this topic as healthy control samples are more readily available than remission samples, which would increase the generalizability of their computational approach.
2. Could the authors also discuss the generalizability of this analytic framework to other single-cell omics technologies?

Reviewer #2 (Comments to the Authors (Required)):

In this manuscript, the authors developed an autoencoder to identify malignant blasts in a mixed population of normal and neoplastic cells, and also to classify them according to their development stage. To generate their model, they analyzed the expression of surface markers via Spectral flow cytometry in a panel of 20 paired samples, collected both at the diagnosis and at relapse/remission, using 5 remission samples to train their model. Their goal was to identify surface markers that are gained, lost or retained in relapsed samples, and to classify the blasts differentiation states based on the expression of the assessed markers. They claim this would be useful for guiding therapy. However, although the dataset analyzed is interesting, there are many shortcomings in the data presented in the manuscript. Major concerns are related to, the use of proper controls, cohort size, clinical and molecular characteristics of the patient population, methodology, results, interpretation and originality/novelty. Below, my comments:

1. There are concerns about how the data in the study represent real world data of pediatric AML. This is probably due to the small cohort size (N=20). First, although pediatric AML occurs throughout childhood there is only one patient below age 2, so this population is underrepresented. Moreover, while in the presented data, the incidence in males is higher than in females (62% males against 38% females), the actual incidence is the same regarding gender (see <https://seer.cancer.gov/>). Furthermore, mutation frequency also shows imbalances that does not correspond the reality. For instance, KMT2A rearrangements, which are less frequent in childhood and adolescence (10-15%) compared to other mutations (PMID: 31925603) is overrepresented in this age group (28%). The small numbers preclude many analyses in the manuscript. The AAML1031 trial has around 500 patients that didn't receive Bortezomib, so increasing the cohort size for this manuscript is

feasible.

2. The authors state that they trained the autoencoder model exclusively with healthy cells. However, they have used samples from patients in remission, i.e. after induction chemotherapy. The biological behavior of cells exposed to genotoxic agents might not be the same as unexposed normal bone marrow (NBM) cells. A post chemo recovery marrow is not the same as a normal marrow. From this reviewer's experience, early in my career I compared protein expression of a cell cycle regulating protein in post chemo remission samples, vs. normal stem cell donor bone marrows. All post recovery samples had the phosphorylated form of the protein and none of the normal marrow samples did. It requires proof that the post chemo recovery marrows are mirrors of truly "normal" marrows. I appreciate that there are not a lot of normal pediatric patients having routine bone marrows, so this would be a difficult sample to obtain. Perhaps normal stem cell transplant donors might suffice. But, since the training system for what is normal is crucial for the development of what standard to use to determine what is a LAIP, to only use post chemo recovery marrow samples invites doubt. Furthermore, the composition of marrow may change with age, especially between infancy, young childhood, later childhood and adolescence. To treat children of all ages as being equal, for purposes of developing the standard for comparison, invites a different set of potential error. Without this knowledge, how can we tell if the errors in classification may be related to this? Finally, developing this standard on just 5 samples seem inadequate. So, I am concerned that the wrong control was used, in inadequate numbers, and without accounting for potential latent variables.

3. In figure 1, the authors use a panel of surface markers to determine the phenotype of cells taken from patients in remission ('healthy cells') and classified those according to the cell type, using already established knowledge. However, when the same markers were used to classify blasts, the authors identified 37 clusters of cells, of which only 16 were defined as blasts and the rest remained unclassified. Moreover, they state that 'For 12 samples, more than 50% of HSPCs/myeloid cells remained unannotated, including the diagnosis and relapse samples of four patients'. This weakens all the downstream conclusions of this study. Furthermore, another major problem is the lack validation. There is a circular argument at work: the system is trained to recognize normal, so a cell that doesn't fit that must be a leukemic blast, ergo the system is better at picking up leukemic blasts. However, there is no consistent proof that the cells that are reclassified are in fact leukemic blasts, and that this system truly identified the leukemic blasts better. Having all that in mind, it would be necessary to use flow cytometry to sort out a reclassified population and analyse the blast phenotypes with other methods (e.g., single cell RNA sequencing, FISH for a molecular marker, etc.). By doing this, the authors would prove that a reclassified cell is in fact a leukemic blast. In addition, correlating these new analyses with the expression of surface markers most likely would be useful to improve their AI model.

4. The authors do not any give information about the criteria for picking samples from AAML1031 trial, except that the all patients belonged to the Arm A (control group), which received the 'standard AML therapy' without the addition of Bortezomib. Moreover, a few other clinical characteristics (e.g. karyotype, cytogenetics, risk group, leukemic burden, etc.) should be shared to better understand the data. In addition, the information in Suppl. Fig. 4 (mutation panel) should be shared as a table together with the other clinical characteristics, rather than as a heatmap. This is crucial information that might provide important insights to some of the conclusions stated in the discussion section.

5. The authors state that patients with KMT2A rearrangements had the greater surface markers changes of all mutations, comparing remission and relapse samples. This is an interesting finding, but not very well explored. As suggested in point 3, gene and/or protein expression analysis would be an option to assess the biological meaning of this finding.

6. There is another issue with the inadequate consideration of how this autoencoder system could or would be used clinically. I do think there is potential utility in an automated flow-based system for providing a more accurate bone marrow differential. The ability to run a slide through an automated system and have it told you how many leukemic blasts are present would be of utility and could greatly speed up analysis of patient material. However, the utility of this is low in cases of obvious disease, such as at the time of diagnosis, or at frank relapse, offering speed, but not changing the disease determination. Instead, this system may be used as an aid to the pathologist, providing a quick read and alerting them of potential disease, or in a remote area without a pathologist, but with flow cytometry capability (this is, however, is a small niche). Another better use would be for MRD detection, if the system is able to accurately detect LAIP cells that would be missed by morphological examination. However, this topic is not investigated at all. It is proposed that this autoencoder system could more accurately identify the presence/absence of a therapeutically targetable cell surface marker (for CART, or BiTE, or another antibody-based therapy). What is not shown is that this methodology does a better job at identifying surface markers compared to conventional flow cytometry. Furthermore, considering a 5% change in the expression of a surface marker between diagnosis and relapse to be significant doesn't seem appropriate, since a change from 85 to 90%, for instance, doesn't change the decision of considering that the marker is present. Likewise, finding that 5% express a marker, when conventional FC picked up 0% is unlikely to move a patient into a group considered eligible for that targeted therapy. Therefore, it would be a small percentage of cases where this system would move the percentage past a crucial threshold and change a patient from ineligible to eligible. Altogether, I think that there are issues regarding practicality and applicability of the findings on this manuscript. Maybe it would be better to try to demonstrate an advantage of the methodology over conventional flow cytometry.

7. Originality and novelty are also major concerns. The existence of LAIPs is old and established. How much utility is there in better defining the actual percentage of these cells in diagnosis or relapse samples? While it has been repeatedly demonstrated that most AML patients relapse from subclones (based on mutation/genetic events) that are present at the time of diagnosis, I have not seen data showing that these sub-clones have different LAIPs at the time of diagnosis. In fact, even if they did, and this

could be identified by this autoencoder system, I don't know how identifying the different LAIP subsets based on surface markers, would help to predict which of those is likely to cause relapse. So, again, I am back to asking what the utility of this is autoencoder system?

Minor comments:

1. Sup. Figure 1C is mistakenly labelled as 1D
2. Please add the legends of the Sup. Figures in a separate page to allow better reading
3. Sup. Figure 4 letters are really small and the data should be presented as a table rather than a heatmap

Responses

Remark: Blast annotations

Reviewer 1 major point 1

“The authors annotate only 16 of 37 non-remission HSPC/myeloid clusters as blasts, labeling only clusters with clear deviations in cluster marker expression from healthy cells. This makes evaluation of the classifier accuracy difficult, as only a fraction of the data has ground truth labels, and that fraction is the most easily identified malignant cells. The authors should use additional tools to annotate a great proportion of these data. The clinical blast counts should give a ballpark estimate of blast numbers in each patient, which could help guide annotation. Further, patient-specific clustering and patient-specific UMAP visualization often enable clearer segregation of healthy and malignant populations, which should enable annotation of more cells. Expanding the training set to include malignant cells with less obvious aberrant phenotypes may improve overall classifier performance and enable more insights in blast characterization in later figures.”

Response

We thank reviewer 1 for their comment and recognize the limitations of the limited annotation. Although we trained the classifier by leaving distinct blast phenotypes out, we admit that more annotated data could be beneficial. Therefore, we used clustered, visualized and annotated non-remission cells per patient. This led us to fully annotate all HSPCs/myeloid cells, including cells that were previously predicted to be healthy. We retrained the classifier using this new annotation. Although the overall accuracy decreased slightly (0.96 to 0.90), we believe that with the additional annotation we capture a more complete blast phenotype. However, the downstream results do not seem to change much.

Actions

Patient-specific clustering and annotation of non-remission cells, followed by retraining of the classifier using more annotated cells. All downstream analyses have been repeated with the new blast prediction.

Remark: Use of healthy bone marrow samples

Reviewer 1 minor point 1

“Could the authors discuss if this analytic framework would be suitable using healthy control samples rather than remission samples as training data? While adding additional samples may be beyond the scope of a revision, it would be useful to have some discussion on this topic as healthy control samples are more readily available than remission samples, which would increase the generalizability of their computational approach.”

Response

We thank the reviewer for this comment and are happy to discuss our decision to use remission samples instead of healthy controls in the revised manuscript (Results section 3.2). Our framework detects cells that deviate from the reference. Getting any bone marrow from pediatric age-matched donors is very difficult as this invasive procedure is usually performed only in children with relevant hematologic abnormalities. Even if no leukemia is diagnosed in the biopsy, there is usually a reason for having a bone marrow evaluation and the sample cannot be considered a “healthy” control sample. True healthy bone marrow samples without an underlying hematological abnormality are often obtained from adults undergoing orthopedic surgery. However, it has been shown that adult bone marrow is different compared to pediatric bone marrow (Proytcheva 2013). Thus, our framework would detect also this deviation, which is not due to the presence of blasts, but due to age. We account for the differences due to age by using remission samples capturing the full age range of our patient cohort (0 to 21 years). We are aware that using remission samples as reference implicates that we consider a bone marrow recovering from chemotherapy as reference. Since it is possible that the chemotoxic treatment had an effect on the bone marrow, remission samples provide the unique ability over healthy samples to account also for this, which is important in relapse samples, where the leukemia reappears on the background of a chemotherapy-treated bone marrow. This would not be possible with healthy bone marrow samples. Since all patients in this study, including those with the remission samples, were treated in the same study arm with the same chemotherapy, it can be assumed that the chemotherapy-induced effects are similar in all patients and using remission samples from this exact cohort provide the best reference and opportunity to identify leukemic blasts.

Actions

We added reasoning for using remission bone marrow samples in the results section 3.2

Remark: Generalizability of analytic framework

Reviewer 1 minor point 2

“Could the authors also discuss the generalizability of this analytic framework to other single-cell omics technologies?”

Response

This is a very interesting suggestion. This method is indeed applicable to other single-cell omics technologies. For any modality with a suitable reference, an anomaly detection framework can be developed. For example, for single-cell mass cytometry, Wagner *et al.* used an autoencoder framework for anomaly detection in breast cancer, assigning a “phenotypic abnormality” score to each single cell (Cell, 20219). Although, to our knowledge, there is no work detecting malignant cells from scRNAseq using a similar workflow as ours. Anomaly detection in scRNAseq is used for

detecting rare cells types or filtering out cells that don't match the celltypes of interest, (Mallick *et al.* Briefings in Bioinformatics, 2023; Feng *et al.* 2020, BioRxiv).

Actions

We expanded the discussion accordingly

Remark: Data representation of real world

Reviewer 2 major point 1

“There are concerns about how the data in the study represent real world data of pediatric AML. This is probably due to the small cohort size (N=20). First, although pediatric AML occurs throughout childhood there is only one patient below age 2, so this population is underrepresented. Moreover, while in the presented data, the incidence in males is higher than in females (62% males against 38% females), the actual incidence is the same regarding gender (see <https://seer.cancer.gov/>). Furthermore, mutation frequency also shows imbalances that does not correspond the reality. For instance, KMT2A rearrangements, which are less frequent in childhood and adolescence (10-15%) compared to other mutations (PMID: 31925603) is overrepresented in this age group (28%). The small numbers preclude many analyses in the manuscript. The AAML1031 trial has around 500 patients that didn't receive Bortezomib, so increasing the cohort size for this manuscript is feasible.”

Response

We appreciate the reviewer's comments about the differences between our cohort of 20 patients and the general pediatric AML population. The reviewer is correct that the differences are explained by the sample size, and the sample size is limited by the availability of a paired sample from relapse. In fact, the COG Myeloid Disease Committee looked at the inventory of banked samples in 2020 and found that the number of cases with a banked relapse sample was only 11% of the number of cases with a banked diagnosis sample. Other requirements for samples to be used for future biology research include having multiple vials banked at both time points and having above a stated threshold of cells in the vial. The actual number of cases that meet all the necessary criteria in the end is rather limited. As the reviewer correctly points out, we have a small number of infants in the cohort, which may reflect additional practical difficulties obtaining sufficient marrow for optional banking from very small patients. While it is unfortunate that our cohort did not include many infants, KMT2A rearrangements are a very common feature of the infant population (Bolouri *et al.*, *Nat Med*, 2018), and this cytogenetic category is well represented. Although we would be eager to analyze more pairs from AAML1031 Arm A, the number of additional samples available in reality will be incremental.

We added a sentence in the text to clarify that our data does not represent AML incidence, nor should be used to draw conclusions on the prevalence of age, sex, race or genetic mutations based on this dataset. However, we do investigate in how many patients each marker changes from diagnosis to relapse. We also added to the discussion that these observations need to be validated in bigger cohorts in the future, to better represent the full range of pediatric AML cases.

Actions

- We added a section in the methods describing age and mutation frequency differences
- Where we discuss markers with therapeutic potential based on for how many patients the marker changes, we now added that our cohort is not fully representative and that these markers should be validated in other cohorts.

Reviewer 2 major point 2

“The authors state that they trained the autoencoder model exclusively with healthy cells. However, they have used samples from patients in remission, i.e. after induction chemotherapy. The biological behavior of cells exposed to genotoxic agents might not be the same as unexposed normal bone marrow (NBM) cells. A post chemo recovery marrow is not the same as a normal marrow. From this reviewer's experience, early in my career I compared protein expression of a cell cycle regulating protein in post chemo remission samples, vs. normal stem cell donor bone marrows. All post recovery samples had the phosphorylated form of the protein and none of the normal marrow samples did. It requires proof that the post chemo recovery marrows are mirrors of truly "normal" marrows.

I appreciate that there are not a lot of normal pediatric patients having routine bone marrows, so this would be a difficult sample to obtain. Perhaps normal stem cell transplant donors might suffice. But, since the training system for what is normal is crucial for the development of what standard to use to determine what is a LAIP, to only use post chemo recovery marrow samples invites doubt.

Furthermore, the composition of marrow may change with age, especially between infancy, young childhood, later childhood and adolescence. To treat children of all ages as being equal, for purposes of developing the standard for comparison, invites a different set of potential error. Without this knowledge, how can we tell if the errors in classification may be related to this? Finally, developing this standard on just 5 samples seem inadequate. So, I am concerned that the wrong control was used, in inadequate numbers, and without accounting for potential latent variables. “

Response

We agree with the reviewer's comment about differences between post-chemo and untreated bone marrow. However, we think that precisely these differences make remission samples a suitable control rather than untreated bone marrow. The framework detects cells that deviate from the reference. In our case, we want this deviation to be due to the malignancy of the cell. Therefore, using healthy bone marrow as reference would cause the framework in relapse samples to also detect cells that are different due to chemotherapy exposure. Remission cells in this case are therefore better suited as a reference than healthy cells. One could, with sufficient healthy, remission and malignant cells use quite a different machine learning set up to detect differences between healthy and malignant cells that are not different between healthy and remission cells. However, this is another question and framework entirely.

As pointed out, healthy pediatric bone marrow samples are scarce as this invasive procedure is not lightly performed on children. Using bone marrow from routine bone marrow donors, as reviewer 2 suggests, leads to differences in the bone marrow sample due to age as these donors are often adults (as also pointed out by reviewer 2), again capturing different causes of deviation from the reference.

Regarding the age of patients in our cohort, we agree that the bone marrow undergoes changes with age, including the age-range included in our study (0-21 years). However, the five remission samples that we have include samples from both the youngest and oldest patients (0 and 21 years) and two patients in the early childhood (5 and 6 years). This covers all age categories mentioned by Proytcheva (2013), that are also found in our cohort.

Actions

We added reasoning for using remission bone marrow samples as reference in the results section 3.2

Reviewer 2 major point 3

"In figure 1, the authors use a panel of surface markers to determine the phenotype of cells taken from patients in remission ('healthy cells') and classified those according to the cell type, using already established knowledge. However, when the same markers were used to classify blasts, the authors identified 37 clusters of cells, of which only 16 were defined as blasts and the rest remained unclassified. Moreover, they state that 'For 12 samples, more than 50% of HSPCs/myeloid cells remained unannotated, including the diagnosis and relapse samples of four patients'. This weakens all the downstream conclusions of this study.

Furthermore, another major problem is the lack of validation. There is a circular argument at work: the system is trained to recognize normal, so a cell that doesn't fit that must be a leukemic blast, ergo the system is better at picking up leukemic blasts. However, there is no consistent proof that the cells that are reclassified are in fact leukemic blasts, and that this system truly identified the leukemic blasts better. Having all that in mind, it would be necessary to use flow cytometry to sort out a reclassified population and analyse the blast phenotypes with other methods (e.g., single cell RNA sequencing, FISH for a molecular marker, etc.). By doing this, the authors would prove that a reclassified cell is in fact a leukemic blast. In addition, correlating these new analyses with the expression of surface markers most likely would be useful to improve their AI model."

Response

The reviewer makes two points in this comment. First, that the abundance of unannotated cells weakens the downstream conclusions. To remedy this, we've used patient-specific clustering of non-remission cells to annotate all cells. Using this new and more complete annotation, we retrained our classifier to predict blasts on single cell level. Although the overall accuracy decreased slightly (0.96 to 0.90), we believe this is due to the increase in difficulty. Since, with the additional annotation we capture a more complete blast phenotype, we also include less

aberrant phenotypes, making classification harder. However, we would like to emphasize that the downstream results and findings did not change.

Secondly, the reviewer points out the lack of validation, which undermines the claim that our method is better at picking up blasts. However, we do not claim to be better at identifying leukemic blasts. Although we can imagine there is a benefit to single-cell classification of blasts, rather than cluster-level annotation, we have not shown this benefit. We do believe our method scales better than manual annotation, where dedicated time of an expert is needed to annotate. Additionally, this approach is more consistent (once trained) than manual annotation, as it doesn't differ between annotators or clustering methods.

Actions

We performed patient-specific clustering and annotation of non-remission cells, followed by retraining of the classifier using more annotated cells. All downstream analyses have been repeated with the new blast prediction.

Remark: patient sample selection

Reviewer 2 major point 4

“The authors do not give information about the criteria for picking samples from AAML1031 trial, except that all patients belonged to the Arm A (control group), which received the 'standard AML therapy' without the addition of Bortezomib. Moreover, a few other clinical characteristics (e.g. karyotype, cytogenetics, risk group, leukemic burden, etc.) should be shared to better understand the data. In addition, the information in Suppl. Fig. 4 (mutation panel) should be shared as a table together with the other clinical characteristics, rather than as a heatmap. This is crucial information that might provide important insights to some of the conclusions stated in the discussion section.”

Response

We added all the requested information to the manuscript and provided the table with the mutational data in a new format. Our criteria for selecting samples included the following: diagnosis/relapse pairs from patients enrolled on AAML1031, Arm A, who consented to banking and who had 3 or more vials of bone marrow (not peripheral blood) banked at both timepoints. The requirement for multiple vials is imposed by the NCI to avoid depletion of the holdings for any individual patient and timepoint. We did not include patients from the other arms to avoid a potential confounding effect of Bortezomib treatment, which could affect the phenotype.

Actions

- Suppl. Fig. 4 is now suppl. Table 7
- We provided more information about sample selection in the methods section 9.1
- We split Suppl. Table 1 into two tables (Suppl. Table 1&2), a demographic table and a clinical data table. The latter now also contains karyotype, risk classification and SCT status.

Remark: Meaning of KMT2A rearrangement surface marker changes

Reviewer 2 major point 5

“The authors state that patients with KMT2A rearrangements had the greater surface markers changes of all mutations, comparing remission and relapse samples. This is an interesting finding, but not very well explored. As suggested in point 3, gene and/or protein expression analysis would be an option to assess the biological meaning of this finding.”

Response

We agree with reviewer 2 that it would be interesting to understand the biological meaning of these surface marker changes in patients with KMT2A rearrangements. However, this this would be an interesting research avenue for the future and beyond the scope of our manuscript which is focused on the computational framework to identify the immunophenotypic composition of AML blasts in an unbiased fashion. Additionally, the community analyzing AML samples could test more markers when having a KMT2A-rearranged sample to validate if a similar trend is observed.

Actions

We clarified the discussion

Remark: Clinical relevance of our method

Reviewer 2 major point 6

“There is another issue with the inadequate consideration of how this autoencoder system could or would be used clinically. I do think there is potential utility in an automated flow-based system for providing a more accurate bone marrow differential. The ability to run a slide through an automated system and have it told you how many leukemic blasts are present would be of utility and could greatly speed up analysis of patient material. However, the utility of this is low in cases of obvious disease, such as at the time of diagnosis, or at frank relapse, offering speed, but not changing the disease determination. Instead, this system may be used as an aid to the pathologist, providing a quick read and alerting them of potential disease, or in a remote area without a pathologist, but with flow cytometry capability (this is, however, is a small niche). Another better use would be for MRD detection, if the system is able to accurately detect LAIP cells that would be missed by morphological examination. However, this topic is not investigated at all. It is proposed that this autoencoder system could more accurately identify the presence/absence of a therapeutically targetable cell surface marker (for CART, or BiTE, or another antibody-based therapy). What is not shown is that this methodology does a better job at identifying surface markers compared to conventional flow cytometry. Furthermore, considering a 5% change in the expression of a surface marker between diagnosis and relapse to be significant doesn't seem appropriate, since a change from 85 to 90%, for instance, doesn't change the decision of considering that the marker is present. Likewise, finding that 5% express a marker, when conventional FC picked up 0% is unlikely to move a patient into a group considered eligible for that targeted therapy. Therefore, it would be a small percentage of cases where this system would move the percentage past a crucial threshold and change a patient from ineligible

to eligible. Altogether, I think that there are issues regarding practicality and applicability of the findings on this manuscript. Maybe it would be better to try to demonstrate an advantage of the methodology over conventional flow cytometry. ”

Response

We acknowledge reviewer’s 2 point of lack of clear clinical applicability. Our tool is currently designed as a post-hoc analysis tool, to facilitate researcher to quickly identify blasts in their dataset. Although not the scope of our manuscript, computational modelling has advanced such that a similar framework could be leveraged in the clinic. Given many datasets with valid reference and disease samples, a powerful model could learn to mitigate batch and patient effects. The model would learn to distinguish the malignant versus healthy signal from the batch and patient signal, if trained on sufficient high-quality data. This would mean that MRD detection or developmental stage composition of unseen samples could be predicted in the clinic.

Additionally, the reviewer points out that we do not show that our method is better at identifying surface markers compared to conventional flow cytometry. However, we do not claim or aim to be better. However, our tool provides a more scalable and fine-grained annotation than manual, cluster level annotation.

The question of what percentage expression is clinically meaningful is an interesting one. One could make that argument that even if a target is expressed on only 5% of AML cells, if that 5% includes the cells that would be responsible for relapse, then it is clinically meaningful. Several current pediatric AML clinical trials incorporating targeted therapies do not use a minimum expression level as inclusion criteria (e.g. AAML1831 (NCT04293562) and gemtuzumab ozogamicin targeting CD33; PEPN2113 (NCT05146739) and uproleselan targeting E-selectin ligand).

Actions

- In the revised manuscript we now show all suggested immunotherapy markers in Figure 4A, where the percentages are shown
- We expanded the discussion on how computational methods could detect differences in the future and the possibility to expand to larger marker panel

Remark: Originality and novelty

Reviewer 2 major point 7

“Originality and novelty are also major concerns. The existence of LAIPs is old and established. How much utility is there in better defining the actual percentage of these cells in diagnosis or relapse samples? While it has been repeatedly demonstrated that most AML patients relapse from subclones (based on mutation/genetic events) that are present at the time of diagnosis, I have not seen data showing that these sub-clones have different LAIPs at the time of diagnosis. In fact, even if they did, and this could be identified by this autoencoder system, I don't know how identifying the different LAIP subsets based on surface markers, would help to predict which of those is likely to cause relapse. So, again, I am back to asking what the utility of this is autoencoder

system?”

Response

We agree with reviewer 2 that the existence of LAIP is established, which is also the basis for our work. Fig. 4A shows that in several patients several markers (e.g. CD34, CD117, CD38, HLA-DR, CD56) are not uniformly expressed in the blasts, indicating that there are blast populations with different LAIP present at diagnosis. However, since we do not have single-cell genetic/mutational events, we cannot infer whether different subclones are responsible for these different LAIPs or whether this heterogeneity arises within one subclone. Additionally, we are not able to predict whether a certain LAIP causes relapse, since we are unable to say with certainty which cells of LAIP causes the relapse. Even if we could identify these cells, making a prediction for novel patients needs a lot training data and another type of model.

The originality and conceptual advance of our work lies in scalable, automated and data-driven blast detection. Additionally, we are able to determine the developmental stage of AML at the single-cell level. The relationship between differentiation stage and drug response in AML is an emerging topic. Our tool can help to investigate if the developmental stage heterogeneity is associated with response to therapy. It is not designed for immediate treatment decisions in the clinic, but rather understanding treatment responses of patients retrospectively. This knowledge could foster the design of studies, which systematically investigate this further and ultimately lead to better informed treatment decisions in the future. Apart from the methodology, we also present a novel finding of an unstable KMT2A blast phenotype. KMT2A rearrangements have a high tendency to relapse both to a more immature or mature developmental stage. Studies which high-resolution measurements of KMT2A-rearranged cells could further investigate the effects or causes of the phenotypic shift and determine the clinical impact.

Actions

We extended the discussion for clarity and pointing out the conceptual advance and novelty of our report

Remark: Mislabeled figure

Reviewer 2 minor point 1

“Sup. Figure 1C is mistakenly labelled as 1D.”

Response

Apologies, we relabeled Suppl. Figure 1C correctly.

Remark: Separate legends of Suppl. Figure

Reviewer 2 minor point 2

“Please add the legends of the Sup. Figures in a separate page to allow better reading.”

Response

We did this in the revised version of the manuscript. We added the figure legends for the supplementary figures separately in the Supplementary Information.

Remark: Suppl. Fig. 4 to table

Reviewer 2 minor point 3

“Sup. Figure 4 letters are really small and the data should be presented as a table rather than a heatmap.”

Response

We have removed suppl. Fig. 4 and added it as suppl. Table 7 in the revised version of the manuscript.

August 5, 2024

RE: Life Science Alliance Manuscript #LSA-2024-02674R

Ms. Alice Driessen
IBM Research - Zurich
Säumerstrasse 4
Rüschlikon 8803
Switzerland

Dear Dr. Driessen,

Thank you for submitting your revised manuscript entitled "Identification of single cell blasts in pediatric Acute Myeloid Leukemia using an Autoencoder". We would be happy to publish your paper in Life Science Alliance pending final revisions necessary to meet our formatting guidelines.

- please consider Reviewer 2's remaining comments. The Discussion addresses some of these points, and mentions that the study is meant to offer a research tool. This should also be mentioned in the Abstract and Introduction to avoid the critiques related to clinical application, since that is not the current aim. Any insight into time saved with this approach would be good to add.
- please be sure that the authorship listing and order is correct
- please upload your main manuscript text as an editable doc file
- please add a Summary Blurb/Alternate Abstract in our system
- please add the Twitter handle of your host institute/organization as well as your own or/and one of the authors in our system
- please add ORCID ID for corresponding (and secondary corresponding) author--you should have received instructions on how to do so
- there seems to be a corresponding author discrepancy between the manuscript and the system. María Rodríguez Martínez and Burkhard Becher are listed as corresponding authors in the manuscript while Alice Driessen and María Rodríguez Martínez are listed on the submission page. Please rectify this discrepancy.
- please label the 'Abstract' (summary) as 'Abstract'
- please consult our manuscript preparation guidelines <https://www.life-science-alliance.org/manuscript-prep> and make sure your manuscript sections are in the correct order
- figure/table captions should be listed at the end of manuscript (both main and supplementary figures/tables)
- please list author contributions for Pavel Sumazin and Richard Aplenc in the manuscript
- contributions listed for Richard Aplenc do not qualify for authorship. Please either update the contributions in our system and in the Author Contributions section of the manuscript, or let us know if the author should be removed.
- please add a callout for Figure 4G and S3A-B to your main manuscript text
- The Methods and the References in the Supplemental file should be incorporated into the corresponding sections of the main manuscript. There are no length restrictions for these sections.
- supplementary tables should be submitted in .xls format
- please enter grant information in the system

A. FINAL FILES:

B. MANUSCRIPT ORGANIZATION AND FORMATTING:

Sincerely,

Reviewer #1 (Comments to the Authors (Required)):

The authors have addressed my concerns; I have no additional comments.

Reviewer #2 (Comments to the Authors (Required)):

In this reviewed manuscript, the authors addressed several points raised in the first version of the manuscript. However, a few responses are rather unsatisfactory and some concerns remains. Below, a few comments:

Remark: Data representation of real world

Major Point 1: the authors successfully addressed this point.

Major Point 2: Although the authors stated that it was difficult to get more samples, the small sample size (N=5) for the control group is still a concern. Therefore, this remains a major limitation of this study.

Major Point 3: the authors re-analyzed the data with a slightly different approach, which is more satisfactory. However, they state the following: "we do not claim to be better at identifying leukemic blasts.". If the autoencoder is not better at identifying blasts, then a major strength is missed.

Remark: patient sample selection

Major Point 4: the authors successfully addressed this point.

Major Point 5: the authors successfully addressed this point.

Remark: Clinical relevance of our method

Major Point 6: The authors claimed that "their tool is designed as a post-hoc analysis tool, to facilitate researcher to quickly identify blasts in their dataset.". If the reason is to more quickly classify specimens, then they need to show how much time is actually saved, since this would have a clinical impact. However, if this autoencoder is designed for research purposes, then saving time during annotation would be a very small use for this tool. This limits the application of this methodology.

Remark: Originality and novelty

Major Point 7: The authors state that their workflow "is not designed for immediate treatment decisions in the clinic, but rather understanding treatment responses of patients retrospectively.". However, if the use of the tool is for research purposes only, then its applicability is very limited and its goal unclear.

Remark: Mislabeled figure

Minor point 1: the authors successfully addressed this point.

Remark: Separate legends of Suppl. Figure

Minor point 2: the authors successfully addressed this point.

Remark: Suppl. Fig. 4 to table

Minor point 3: the authors successfully addressed this point.

August 19, 2024

RE: Life Science Alliance Manuscript #LSA-2024-02674RR

María Rodríguez Martínez
Yale School of Medicine
333 Cedar Street
New Haven, Connecticut 06510

Dear Dr. Rodríguez Martínez,

Thank you for submitting your Methods entitled "Identification of single cell blasts in pediatric Acute Myeloid Leukemia using an Autoencoder". It is a pleasure to let you know that your manuscript is now accepted for publication in Life Science Alliance. Congratulations on this interesting work.

DISTRIBUTION OF MATERIALS:

Again, congratulations on a very nice paper. I hope you found the review process to be constructive and are pleased with how the manuscript was handled editorially. We look forward to future exciting submissions from your lab.

Sincerely,
